# De Novo Transcriptome Sequencing and Analysis of Differential Gene Expression among Various Stages of Tail Regeneration in *Hemidactylus flaviviridis*

**DOI:** 10.3390/jdb10020024

**Published:** 2022-06-14

**Authors:** Sonam Patel, Isha Ranadive, Pranav Buch, Kashmira Khaire, Suresh Balakrishnan

**Affiliations:** Department of Zoology, Faculty of Science, The Maharaja Sayajirao University of Baroda, Vadodara 390002, India; sppatel191@gmail.com (S.P.); isharanadive0404@gmail.com (I.R.); prbuch@gmail.com (P.B.); kashmira.k-zoophd@msubaroda.ac.in (K.K.)

**Keywords:** transcriptome, house gecko, Wnt, FGF, Shh, BMP, early blastema, blastema, differentiation, tail regeneration

## Abstract

Across the animal kingdom, lizards are the only amniotes capable of regenerating their lost tail through epimorphosis. Of the many reptiles, the northern house gecko, *Hemidactylus flaviviridis*, is an excellent model system that is used for understanding the mechanism of epimorphic regeneration. A stage-specific transcriptome profile was generated in the current study following an autotomized tail with the HiSeq2500 platform. The reads obtained from de novo sequencing were filtered and high-quality reads were considered for gene ontology (GO) annotation and pathway analysis. Millions of reads were recorded for each stage upon de novo assembly. Up and down-regulated transcripts were categorized for early blastema (EBL), blastema (BL) and differentiation (DF) stages compared to the normal tail (NT) by differential gene expression analysis. The transcripts from developmentally significant pathways such as FGF, Wnt, Shh and TGF-β/BMP were present during tail regeneration. Additionally, differential expression of transcripts was recorded from biological processes, namely inflammation, cell proliferation, apoptosis and cell migration. Overall, the study reveals the stage-wise transcriptome analysis in conjunction with cellular processes as well as molecular signaling pathways during lizard tail regeneration. The knowledge obtained from the data can be extrapolated to configure regenerative responses in other amniotes, including humans, upon loss of a complex organ.

## 1. Introduction

On the evolutionary scale, reptiles (e.g., geckos) are the closest group to mammals that can restore lost body parts. Moreover, the histological features of reptiles and mammals are highly comparable, making the former an appealing group for studying tissue and organ regeneration [1]. Despite their limitations, lizards can regrow lost parts, similar to their vertebrate counterparts, such as fish and amphibians [2,3,4]. In contrast to the metamerically segmented original tail, lizards replace their lost appendage (tail) with an unsegmented tail [5]. Nonetheless, this replaced tail is good enough for the animal to regain social acceptability and survival. Hence, though neglected, lizards provide an excellent platform to answer the questions related to regeneration biology [6].

Observations in our lab suggest that tail regeneration in the lizard *Hemidactylus flaviviridis* follows the epimorphic regeneration [7]. Epidermal cells from the periphery of the tail migrate to cover the wound surface immediately after autotomy. This wound-healing phase is completed within 48 h of autotomy and is accomplished solely through cell migration rather than cell division. By 4 days post-autotomy (dpa), a flat early blastema is formed covering the stump. The initiation of blastema is accompanied by histolysis of stump tissues such as bone and muscle, from which dedifferentiated cells arise and aggregate directly beneath the early blastema [4]. These cells re-enter the cell cycle and give rise to the blastema, a mass of mesenchyme-derived cells thought to be substantially, if not entirely, formed from 6 dpa dedifferentiation of previously differentiated cells. The blastema is only noticeable as a bit of protuberance during the early bud stage, but when cell division continues, it extends into a cone-stage blastema. The dedifferentiated cells re-differentiate into tail tissues (10 dpa) as the blastema expands, following many of the same patterning mechanisms used during embryonic tail formation [8].

*H. flaviviridis*, the northern house gecko, is an emerging model organism that has contributed to research in the domains of evolution and development, population genetics, reproductive physiology, behavior, and functional morphology. Comprehensive gene expression studies for processes such as gecko tail regeneration have been difficult in the past owing to the lack of resources.

With the introduction of next-generation sequencing technology, vast amounts of genomic data from lesser-used model species may now be collected in a short time and effort [9]. Transcriptome datasets thus generated provide cardinal leads in understanding the molecular basis of various biological processes. Moreover, advancements in this technology have produced datasets with fewer intronic, intragenic and repetitive sequences, otherwise hindering genomic assembly. Transcriptomes thus require fewer computational resources for assembly than whole genomes, can be annotated by comparison with protein sequences from even distantly related species, and have high functional information content [10]. We intended to figure out which genes are involved in different stages of lizard tail regeneration in the current study. In order to accomplish this, we analyzed the transcriptomes of early blastema (EBL), blastema (BL), and differentiation stages (DF) and compared the results to the normal tail. The transcriptome was analyzed with the Illumina HiSeq 2500 platform, which used 100 bp paired end sequencing. The current work is the first to compare the entire collection of gene activation that occurs throughout different stages of *H. flaviviridis* tail regeneration. We discovered many important pathways involved in lizard tail regeneration in a stage-specific manner.

## 2. Materials and Methods

### 2.1. Ethical Statement

Animals were handled as per the guidelines of the ‘‘Committee for the Purpose of Control and Supervision of Experiment on Animals (CPCSEA),’’ Government of India, with Institutional Animal Ethics Committee IAEC approval (MSU-Z/IAEC/15-2017).

### 2.2. Animal Maintenance and Tissue Collection

*Hemidactylus flaviviridis* (Rüppell, 1835) were collected and kept in well-ventilated cages. A photo-period of 12:12 was maintained. A constant room temperature and humidity of 36 ± 2 °C and 60–70%, respectively, were maintained during the experimental regime. Animals were acclimatized for at least seven days before the commencement of the study. Only healthy animals with normal intact tails were selected for further procedures. For collecting the tail tissue, lizards were placed on a sterilized tile, followed by inducing autotomy by applying pressure on the segment of tail to be collected. Tail tissues (50 mg) were collected in 1 mL of TRIzol reagent for the following time points: normal tail (0 dpa), early blastema (4 dpa), blastema (6 dpa), and differentiation (10 dpa). The 0 dpa is the normal tail, followed by the early blastema stage, wherein a smooth covering can be seen at the autotomy plane on 4 dpa. After this stage, a conical structure is visible at 6 dpa, which we call the blastema stage. Lastly, at 10 dpa, a long-elongated cone-like structure can be seen, which marks the beginning of the differentiation stage [8].

### 2.3. RNA Extraction, cDNA Library Construction/Preparation

The total RNA was isolated from 50 mg tail tissue using the TRIzol reagent, as reported previously by Ranadive et al. (2018) [4]. The quality and quantity of the isolated RNA were estimated using Qubit Fluorimeter 3.0 and Agilent Tapestation 2200. Then, 1 μg of isolated RNA was used to prepare the transcriptome library using Truseq RNA Library prep kit V2 (Illumina Inc., San Diego, CA, USA, Cat# RS-122-2001) as per the manufacturer’s protocol. The libraries were sequenced on Illumina HiSeq 2500 platform in 100 bp paired end sequencing.

### 2.4. Removal of Sequence and Quality Filtering

Before assembly and mapping, we applied filters to remove low quality reads and trim the 3′ and 5′-end adapters. A trimmomatic program (version 0.36) was used to trim the adapters. FASTQC (version 0.11.5) program was used to assess the quality of the reads. Any read with a Phred score (Q) below 30 was discarded. Further, reads that did not match to a minimum of 50 bp (out of 150 bp sequencing length) were not considered high-quality reads and were also discarded [11].

### 2.5. De Novo Sequencing and Transcriptome Assembly

All the forward reads of all samples were concatenated together. Similarly, the reverse reads of all the samples were concatenated together. This was done to generate a reference assembly that can be used later for analyzing differential gene expression. This single paired end data was used as input to TRINITY for de novo transcriptome assembly. Trinity (version 2.8.4) was run with its default value of 25 using the k-mer algorithm. It helps for the de novo reconstruction of transcriptomes from RNA-seq data. CD-hit (version 4.7, with default parameters) was used to cluster the assembled transcripts and remove redundancy. The resultant transcripts (contigs) were used for further analysis. The entire transcriptome data has been uploaded on NCBI and the accession number of the bio-project is PRJNA522668 (https://www.ncbi.nlm.nih.gov/bioproject/522668; accessed on 15 February 2019).

### 2.6. Functional Annotation and Classification of Transcripts

Annotation of the TRINITY assembled contigs was done by doing a protein blast. Chordata protein database was downloaded from UniProt. Diamond program (version 0.9.24) was used to do a local BLAST (BLASTx with an e-value cut-off of 1 ×10^−10^ and max-targets as 10). The same UniProt database was used to retrieve gene ontology (GO) terms each for biological process, cellular composition and molecular function categories. The top hits from BLAST results for each transcript were taken for further analysis.

### 2.7. Differential Gene Expression Analysis

EdgeR program was used for differential expression estimation. The package EdgeR provides methods to test for differential expression using negative binomial generalized linear models. The dispersion value was taken at 0.1. It uses these normalized count values to calculate differential expression. The p.adj values were computed from the *p*-values using the Benjamini–Hochberg procedure. Differentially expressed genes were decided by a set of cut-offs, which were log2 fold change of ±1. (*p* ≤ 0.05 and FDR ≤ 0.01).

### 2.8. Pathway Analysis

BLAST results with *Mus musculus* were extracted and the top hits of each transcript were chosen for enrichment in KEGG pathways. R package ClusterProfiler (version 3.10.1) was used to do GO enrichment as well as KEGG pathway analysis of all the differentially expressed genes. Pathview package was used to generate KEGG pathway images highlighting the up- and down-regulated genes on the plot. 

Enrichment analysis was performed for the differentially expressed genes with log2 fold change of >1 and <−1 with a p-adj (FDR) value of 0.05.

### 2.9. Statistical Analysis

The data is represented as mean ± S.E.M. One-way analysis of variance (ANOVA) was applied to analyze significant variation in relative fold change of mRNA expression for each gene during different stages of regeneration. Newman–Keuls multiple range test was used to compare the means. A *p* value less than or equal to 0.05 was considered statistically significant.

## 3. Results

### 3.1. Illumina Paired-End Sequencing and De Novo Transcriptome Assembly

RNA sequencing performed using Illumina HiSeq 2500 from tissue samples of lizard tail at 0 dpa, 4 dpa, 6 dpa and 10 dpa generated 100 bp reads for both ends of cDNA. The data yielded 114.96 million (11.50 Gb) raw reads for the normal tail, 84.04 million (8.40 Gb) for the wound healing stage, 83.48 (8.35 Gb) million for blastema stages and 91.66 million (9.17 Gb) for differentiation stage. These raw reads were subjected to adapter trimming and removal of low-quality reads to obtain clean reads. Table 1 depicts the clean read summary obtained for each sample. The clean reads were annotated and a total of 310,553 transcripts were obtained from all the stages of regeneration; 310,553 transcripts were subjected to UniGene clustering to identify the true transcripts. This analysis identified 191,330 true transcripts, which were considered in downstream analyses. In total, 33,960 transcripts out of 191,330 true transcripts got annotated, with great confidence, using the homology search methods.

### 3.2. Functional Annotation and Gene Ontology (GO) Classification

The assembled transcripts were matched, with Chordata protein from UniProt database, using the BLASTX program. A total of 89,899 transcripts were obtained, out of which 55,344 transcripts could find a homolog from the UniProt database. Transcripts that could establish homology relationship, with E-value ≤ 10^−5^ and similarity score ≥ 40%, were retained in the annotation pipeline for further annotation while the others remain un-annotated. Overall, we found that 53,201 of the assembled transcripts had at least one significant hit in the UniProt Chordata database, whereas 34,555 transcripts did not show a significant hit. In this analysis using BLASTX, around 60% of the transcripts could identify a homolog with a minimum E value confidence of 1 ×10^−5^. This indicates the high level of conservation of the proteins. Around 81% of the transcripts could identify a homolog with more than 60% similarity at the protein level with the existing proteins. The distribution of similarity scores from BLASTX results is shown in Figure 1. The top BLASTX hit of each transcript was studied and the organism’s name was extracted. Among the annotated UniGenes, 19,633 transcripts received the top hit with *Anolis carolinensis* (green anole lizard), followed by other reptiles. The ten most frequently appearing organisms among the source organisms of the homologs are shown in Figure 2. 

Further, the gene ontology assignment program was used to categorize the annotated UniGenes functionally. The program could map 11,219 GO terms in total, wherein 7339, 2665 and 1215 were categorized into biological processes (BP), molecular functions (MF) and cellular components (CC), respectively. A few GO terms from each category are shown in Figure 3. The detailed data on each GO term with the number of transcripts found for each GO term can be found in Appendix A.

For BP, the majority of the transcripts were found to be from the regulation of transcription (GO:0006355, 2200) and signal transduction (GO:0007165, 733) (Figure 3A). In the case of MF, 8275 of the transcripts belonged to the nucleic acid binding category (GO:0003676), including its child terms like DNA binding (GO:0003677), RNA binding (GO:0003723) and transcription factor activity (GO:0003700) (Figure 3B). Figure 3C shows that most of the assignments were given to transcripts from the integral component (GO:0016021, 7593) followed by the nucleus (GO:0005634, 4251) (Figure 3C).

### 3.3. Differential Gene Expression Analysis

The differentially expressed genes for EBL, BL and DF stages were identified with respect to the normal tail as reference (Figure 4). The level of significance of this difference was checked using the *p*-value (with a cut off of 0.05) from the hypothesis testing. The results of these analyses are shown in Table 2. Further, these differentially expressed transcripts were categorized into up- and down-regulated transcript numbers for each subcategory of BP, MF and CC (Figure 5). A substantial increase in the number of up-regulated transcripts was noted for immune response and inflammatory response in the EBL stage compared to the BL stage in comparison to the normal tail. However, the numbers of up-regulated transcripts for immune and inflammatory responses were again increased for the DF stage with respect to NT (Figure 5A). The number of down-regulated transcripts exhibited exactly opposite trends for all three stages (Figure 5B). Interestingly, the number of up-regulated transcripts for regulation of transcription and signal transduction was found to be more for all three stages, but the highest in the BL stage, being a highly proliferative phase in comparison to NT (Figure 5A).

Like BP, most of the subcategories of MF, especially nucleic acid binding, DNA binding and RNA binding displayed a similar pattern for up-regulated transcripts with the highest number in the BL stage followed by a decrease in DF and EBL stages compared to NT (Figure 5C). Interestingly, the number of down-regulated transcripts for protein serine/threonine kinase activity was found to be more during the BL stage than in the EBL or DF stage (Figure 5D). In the case of CC, the number of up-regulated transcripts for the integral component of the membrane was found to be almost similar and high for both EBL and BL stages but drastically reduced for DF in comparison to NT (Figure 5E). The prominent presence of down-regulated transcripts for extracellular matrix was noted in EBL and DF stages but not in the BL stage (Figure 5F).

### 3.4. Pathway Analysis

From the heatmap of differentially expressed genes (DEGs), genes that showed significant changes across all the stages of regeneration were identified. Most of these genes were found to be part of biological processes and signaling pathways. A set of pathways was analyzed thoroughly to understand the level of gene expression regulation in the tissues at all stages of regeneration. Primarily, genes from processes such as angiogenesis, apoptosis and inflammation were reported in this study (Figure 6). Genes from cell signaling pathways such as Shh, FGF, TGF and Wnt were also analyzed. About 85 genes were up-regulated and 145 others were down-regulated at all stages. The expression levels of genes in these pathways are shown in Figure 6.

## 4. Discussion

Vertebrate regeneration has been studied extensively in animal models to understand the underlying mechanism at various levels. Genomic and proteomic profiles from zebrafish and other vertebrates have already been reported [12,13,14,15]. One such attempt was made in the current study wherein comparative analysis of the tail transcriptome from different stages of regeneration in the northern house gecko, *H. flaviviridis*, was studied. The stage-wise transcriptome profile revealed several genes involved in structural and functional aspects of lizard tail regeneration across the stages. Previously, we studied the proteomic profile of regenerating tail in *H. flaviviridis,* wherein differential protein levels of certain growth factors, inflammatory mediators and ECM remodelers were reported [16].

The de novo transcriptome study could map thousands of transcripts from the regenerating tail of the house gecko. Upon differential gene expression analysis, it was noted that the transcripts involved in the regulation of immune response and inflammation were up-regulated at the EBL stage. Similar observations were made during scar-free wound healing of the tail in the green anole lizard (*Anolis carolinensis*) by Xu et al. (2020) [17]. Along the same lines, Vitulo and coworkers compared the transcript levels of molecules regulating events such as metabolism, ECM remodeling, cytoskeletal patterning, etc., in regenerating and scarring appendages of *Anolis carolensis* [18]. However, the present study highlights the micro alterations in the gene expression of molecules that lead to the temporal changes in the microenvironment of the regenerating tail, eventually causing stepwise progress in the form of early blastema, blastema, and differentiated tissue. Further, the pathway analysis revealed decreased levels of the pro-inflammatory cytokine TNF-α at the EBL stage, suggesting resolution of the inflammatory phase. This corroborates our previous results in *H. flaviviridis* upon scarred vs. scar-free wound healing comparison [19]. Similar results have been observed for the *Anolis* model, wherein gene expression of inflammatory mediators drops significantly during blastema formation, illustrating its immunocompromised state [20]. 

TRADD transcript levels were also found to follow the same trend as TNF-α. TRADD has been reported to be essential for governing and regulating TNF-α mediated immune and inflammatory responses [21]. As the regeneration progresses, the levels of these pro-inflammatory mediators decline to pave the way for the proliferation and growth of the healing tail tissue. This can be witnessed by the increased number of transcripts belonging to transcription and signal transduction regulation. Similar observations were made by Nagumantri et al., wherein the levels of genes and proteins involved in inflammation were found to be decreased at 5 dpa during tail regeneration in *H. frenatus* [22]. 

On the other hand, Murawala and colleagues presented the global proteome results for normal tail, early blastema, blastema, and differentiation stages in *H. flaviviridis* [16]. The focus of the above study was mainly on the change in expression of the structural proteins and enzymes involved in the metabolism. However, the reported trend of protein expression for FGF10 and TNF-α matches the present transcriptome result. Amongst the stages, the blastema stage outcompeted the other two for the highest levels of transcripts involved in proliferation and transcription. Transcripts of the major pathways governing these processes such as FGF, Shh, Wnt and TGF were also present during the transcriptome analysis. These signaling pathways control all the cellular and molecular processes required for successful regeneration in zebrafish, [23,24,25], axolotl [14,26], *A. carolinensis* [17,26] and *H. flaviviridis* [4,7,8].

Ranadive et al. (2018) [4] showed that the proliferation of epithelial cells during the early regeneration phases is crucial for scar-free wound healing in *H. flaviviridis*. Fibroblasts are among the first to reach the autotomized site to form the wound epithelium [17]. The presence of FGF family members during the wound healing stage has been reported in various animal models of regeneration [27,28]. In the current study, we noticed differential levels of FGF receptors and ligands across the regeneration stages. The prominent presence of FGF21 at the EBL stage depicts its involvement in cell migration and tissue repair. FGF21 has been reported to enhance angiogenesis and migration of human brain microvascular endothelial cells showing its therapeutic potential for treating human brain injury [29]. Additionally, high levels of PCNA at the EBL stage support the proliferation of cells at the wound site. The success of scar-free wound healing lies in the balance between cell proliferation and apoptosis. Therefore, a decrease in the number of transcripts associated with apoptosis at EBL and BL stages seems to pave the way for the growth of the autotomized tail. Differential expression of genes and proteins involved in cell proliferation and cell cycle regulation have also been reported in other lizards during tail regeneration [17,20].

Besides FGF, Wnt signaling plays a pivotal role in lizard tail regeneration [7,17]. Varied levels of wnt11, wnt1, wnt4 and wnt6 transcripts were present across the regeneration stages. Most members of Frizzled receptors were also found to be expressed in all three stages during tail regeneration. Xu et al. (2020) [17] showed that Wnt family ligands and receptors were highly expressed at 5 dpa during tail regeneration in the green anole lizard. Wnt/β-catenin signaling has been shown to regulate cell proliferation in distal blastemal cells as well as tissue patterning during caudal fin regeneration in zebrafish [30]. Wnt and BMP signaling controls the formation of the apical epithelial cap during axolotl and zebrafish regeneration [30,31].

Regeneration is a complex process involving a variety of signals as well as cell sources for the regrowth of the lost tissue. Satellite cells residing in adult skeletal muscles could be a potential contributor during regeneration. Exogenous addition of BMP to in vitro cultures of satellite cells has been reported to differentiate into cartilage. Therefore, the presence of TGF-β/BMP pathway members, including the downstream regulators (SMADs), could play a similar role during lizard tail regeneration. TGF-β/BMP signaling governs the differentiation of skeletal elements (bone and cartilage), hence its occurrence during tail regeneration is not surprising.

Crosstalk between signaling pathways suggests that interplay between molecular cues is central to regeneration. However, the spatio-temporal expression of each signaling molecule is essential for the progress of regenerating tissue. Shh and bmp2b signaling play an important role in bone patterning during caudal fin regeneration in zebrafish. Inhibition of the Shh pathway with cyclopamine during fin regeneration impeded cell proliferation as well as fin outgrowth [31]. In the current study, differential levels of shh, ptch1, smo, gli1, gli2 and gli3 transcripts depict the complex network of signal transduction pathways in conjunction with FGF, BMP and Wnt pathways. Overall, the current transcriptome data reveals the stage-specific occurrence of cellular and molecular events. However, future studies with uniquely expressed transcripts for each stage will illuminate the delicate regulatory mechanism beneath the epimorphic regeneration in this amniote model. Lizards elicit a different mode of regeneration compared to other models such as salamanders and zebrafish, and therefore this study provides new insight into the overall regenerative operations. This unique amniote model might have conversed genetic similarities that can be explored in the field of regenerative medicine in order to study tissue repair and regeneration.

## Figures and Tables

**Figure 1 jdb-10-00024-f001:**
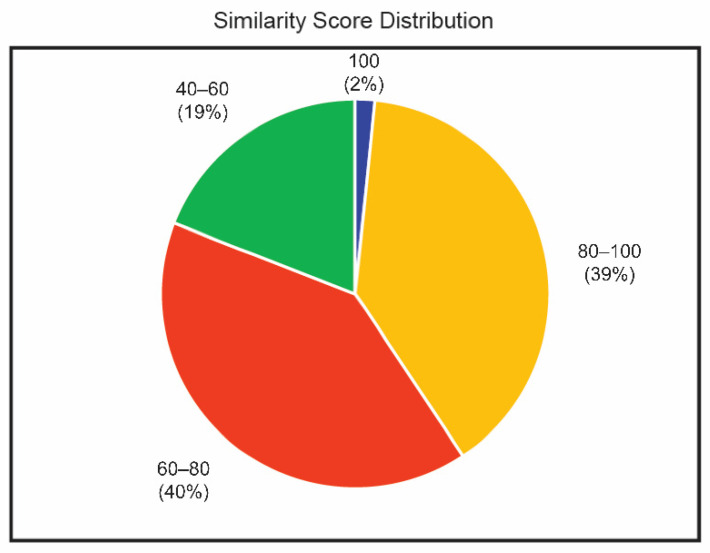
BLASTX similarity score distribution of transcripts.

**Figure 2 jdb-10-00024-f002:**
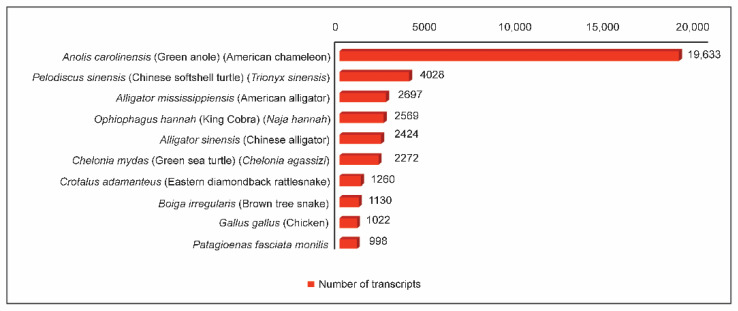
Top ten organisms and the number of transcripts from them in BLASTX hits.

**Figure 3 jdb-10-00024-f003:**
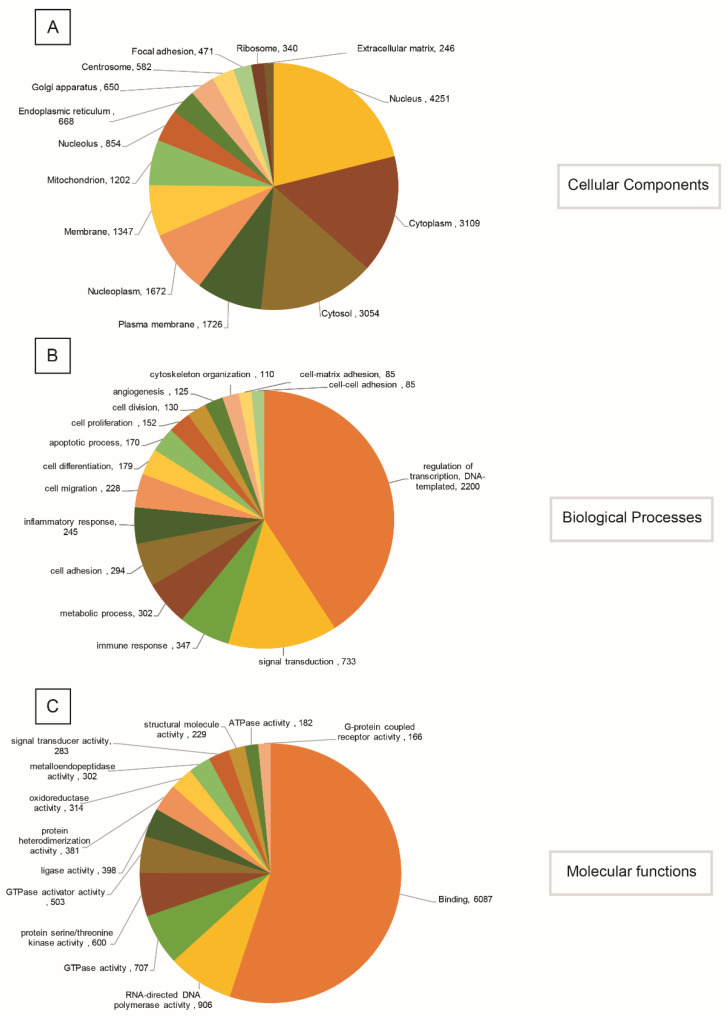
Gene Ontology classification of Unigenes derived from sequencing in *H. flaviviridis* during tail regeneration belonging to categories for cellular components (**A**), biological processes (**B**), and molecular functions (**C**).

**Figure 4 jdb-10-00024-f004:**
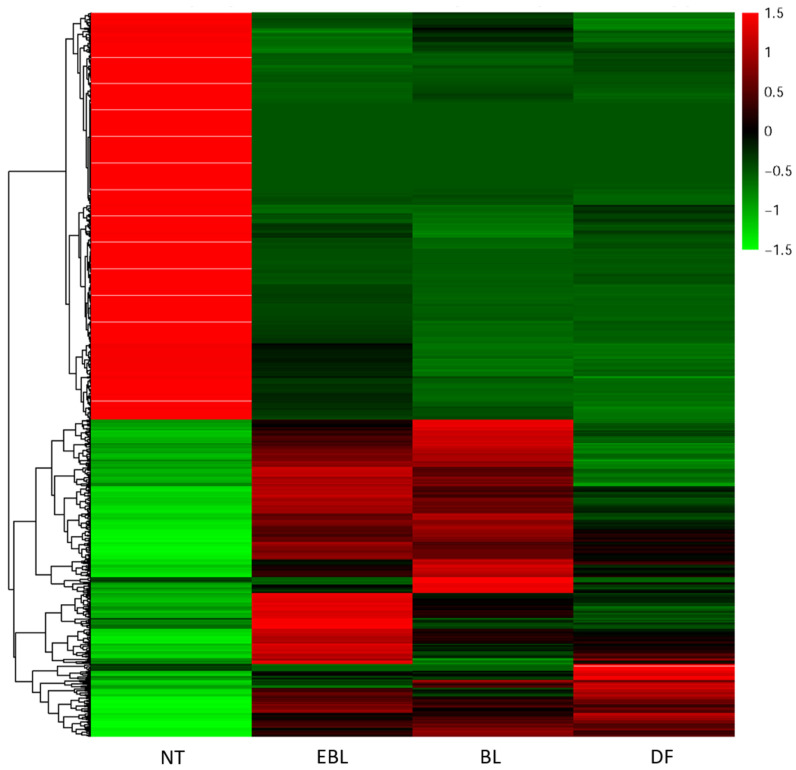
Heat map of differentially expressed transcripts across all the stages (NT: Normal Tail; EBL: Early Blastema; BL: Blastema; DF: Differentiation) of regeneration in *H. flaviviridis*. In the heat map, for normal tail, black bands represent the basal level of expression. The gradient of green and red represents the extent of downregulation or upregulation, respectively, compared to the housekeeping gene. However, for EBL, BL and DF the black, green and red represent basal, and extent of downregulation or upregulation, respectively, compared to normal tail.

**Figure 5 jdb-10-00024-f005:**
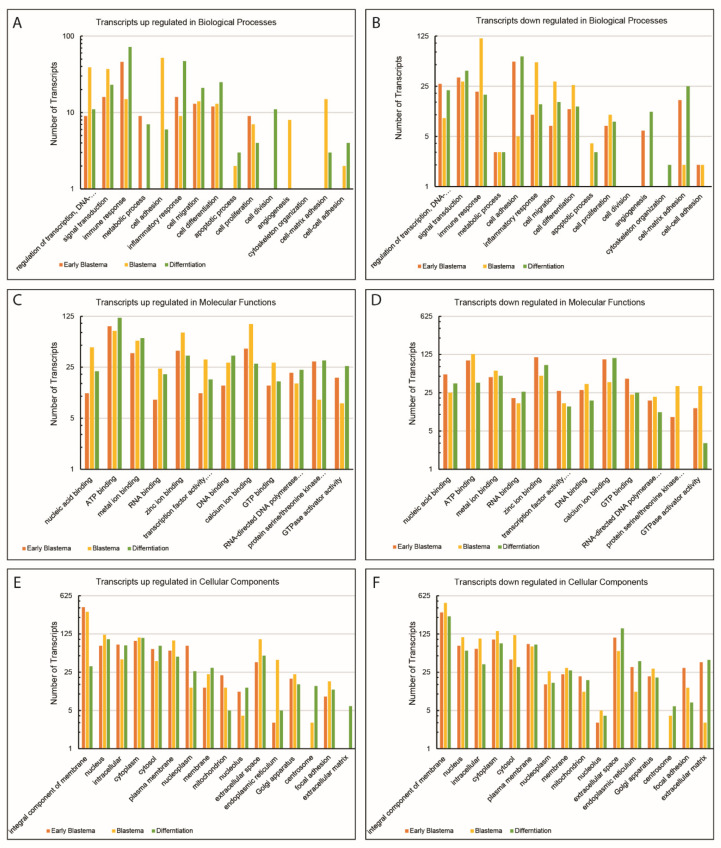
Histogram representation of the gene ontology classification of up- and down-regulated transcripts from early blastema (EBL), blastema (BL) and differentiation (DF) stages with respect to the normal tail during regeneration in *H. flaviviridis.* (**A**,**B**) Up-regulated and down-regulated transcripts in biological processes; (**C**,**D**) Up-regulated and down-regulated transcripts in molecular processes; (**E**,**F**) Up-regulated and down-regulated transcripts in cellular processes.

**Figure 6 jdb-10-00024-f006:**
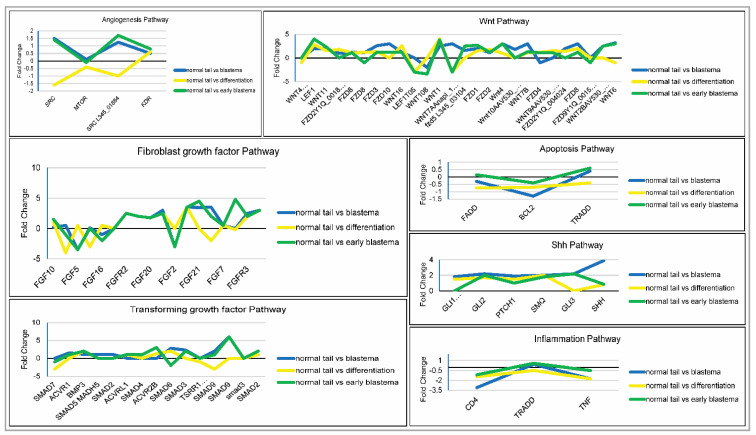
Expression levels of transcripts from developmentally significant signaling pathways in early blastema (EBL), blastema (BL) and differentiation (DF) stages with respect to the normal tail during regeneration in *H. flaviviridis.* The line through zero represents the respective gene expression levels in normal tail samples considered as a reference. Fold changes in gene expression in other samples (EBL, BL and DF) are compared with this reference.

**Table 1 jdb-10-00024-t001:** Clean read summary.

Sample	dpa	Number of Reads (Million)	Number of Bases (Gb)
Normal Tail (NT)	0	106.57	8.81
Early Blastema (EBL)	4	80.48	6.67
Blastema (BL)	6	80.19	6.65
Differentiation (DF)	10	87.37	7.23

**Table 2 jdb-10-00024-t002:** Count of differentially expressed transcripts.

Stages Compared	Transcripts	
	Down-Regulated	Up-Regulated
Normal Tail vs. Early Blastema	922	1282
Normal Tail vs. Blastema	1152	1160
Normal Tail vs. Differentiation	1194	719

## Data Availability

The data presented in this study are openly available in https://www.ncbi.nlm.nih.gov/bioproject/522668, accessed on 1 April 2022.

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
