# Peer review of "De Novo Transcriptome Sequencing and Analysis of Differential Gene Expression among Various Stages of Tail Regeneration in *Hemidactylus flaviviridis"

_jdb, 2022, doi:10.3390/jdb10020024_

Round 1
Reviewer 1 Report
The present manuscript by Patel et al. reports on a de novo transcriptome sequencing and analysis of differential gene expression during various stages of tail regeneration in the lizard Hemidactylus flaviviridis. Epimorphic regeneration results in the regrowth of lost tissues and structures from a population of undifferentiated and proliferating cells known as a blastema. Among amniotes the most impressive example of epimorphic regeneration comes from tail regenerating lizards. Although the phenomenon of tail regeneration has been known for a long time, details of the sequence of molecular-level events are still poorly studied. As such, it would be of interest to researchers studying regeneration, development and related topics. Generally, this study is technically solid, and represents a valuable contribution to our understanding of regenerative processes. Nevertheless, there are a few issues to consider prior to the publication of this work:
1) The authors need to make their data more accessible.
All the most important primary data (Differentially expressed genes, GO, KEGG pathway analysis) should be presented as supplementary data. The Supp. File no 1. with detailed data on each GO term is not available to me.
2) There is a lack of validation of the transcriptomic analysis. It would be helpful to present RT-PCR data for a few key up and down regulated genes.
3) The discussion is generally very useful, highlighting similarities and differences between vertebrate models. A few edits would help to improve this section even more:
The work can be of particular value in terms of comparison if it coherently summarizes various reports. Moreover, the conceptual interpretation of the results is overly narrow in parts.
For example, the authors have published previous articles related to proteome of regeneration tail in this animal. It would be very useful to compare the data of proteome and transcriptome analysis at least for a number of genes. In addition, the data obtained by authors on Hemidactylus flaviviridis should be discussed in comparison with data on a similar model of lizards of another species from the same genus. See: Nagumantri, S.P., Banu, S. & Idris, M.M. Transcriptomic and proteomic analysis of Hemidactylus frenatus during initial stages of tail regeneration. Sci Rep 11, 3675 (2021). https://doi.org/10.1038/s41598-021-83283-0. It would also be very useful to discuss the change in the expression of molecular factors that may be involved in interactions between the wound epidermis and underlying stump tissues, those required in a structure regeneration in vertebrates.
4) The title of the article suggests a special focus on the analysis of Hedgehog signaling , however, both in the results and in the discussions, this type of signaling is of minimal interest.
Author Response
- We thank the reviewer for the constructive suggestion, and we have included the GO term data as a supplementary file.
-
We sincerely take the comment mentioned above, but we are afraid that we might not be able to add the qRT-PCR data for the key up and down-regulated genes at this moment. The primary reason for not adding the validation data in the current manuscript is that we are currently studying the genes involved in the molecular signaling pathways of developmental significance as a separate objective. Hence, we apologize for not including the supporting data at this moment
-
We thank the reviewer for drawing our attention to the particular reference. The reference has been included and discussed in the revised manuscript.
-
We apologize for the wrong incorporation of the title with Shh signaling in the focus. We are still validating the signaling pathways at protein and RNA levels hence the title has been appropriated in the revised manuscript (De novo transcriptome sequencing and analysis of differential gene expression among various stages of tail regeneration in Hemidactylus flaviviridis. We are not focusing on Shh signaling in the current manuscript. Thank you very much for your critical observation.
Reviewer 2 Report
In this study, Patel et al utilized amputated Hemidactylus flaviviridis tail as a model to study the molecular signals involved in regeneration. They described the transcriptome data of the regenerated tail at different stage during tail regeneration. There are several critical issues needed to be addressed in this manuscript.
- The title of this manuscript should be reconsidered. The authors addressed the words as ‘analysis of Shh pathway’ in the title. However, Shh pathway was only described as other pathways in this manuscript. No further experiments were conducted addressed in Shh pathway. Therefore, it is inappropriate to specially addressed analysis of Shh pathway in the title.
- In section of Materials and Methods, how the tissues were prepared should be clearly described. This is the essential information for the definition of the different stages termed in this manuscript, which should also be clearly described.
- In figure 2, the figure legend said that this figure showed top 15 organisms and the number of transcripts from them in BLASTX hits. However, only 10 organisms were presented. In addition, the top 15 organism did not include another gecko that might be the closed species with the transcription data available in public database. Please confirm this result.
- In figure 6, the molecules showed in the TNF related pathway are Smad and BMP?
Author Response
- We apologize for the wrong incorporation of the title with Shh signaling in focus. We are still validating the signaling pathways at protein and RNA levels; hence, we revised the title to “De novo transcriptome sequencing and analysis of differential gene expression among various stages of tail regeneration in Hemidactylus flaviviridis.” We are not focusing on Shh signaling in the current manuscript.
-
The aforementioned suggestions have been incorporated in the revised manuscript.
-
We thank the reviewer for pointing out the details to us. The figure legend has been corrected in the revised manuscript.
- Figure 6 has Tumor Growth Factor pathway (TGF) and components are Smad and BMP
Reviewer 3 Report
This manuscript begs for additional experiments showing validation of
some of the genes found to be up and down regulated. For example, validation via in situ hybridization would strengthen claims about some of these genes.
There are minor editing issues that need to be changed. First and foremost, captions need to match the abbreviations used in Figures 3 and 5. Small editing mistakes such as spaces missing between words are also present. Words can be removed from some sentences to make the English sound a bit clearer.
Author Response
- We highly appreciate the suggestion of validation with other methods. However, we apologize for not being able to include the data in the current manuscript. We are still validating the signaling pathways at protein and RNA levels using both molecular and histological approaches.
-
We thank the reviewer for all the constructive comments. We have corrected the errors at relevant places in the entire manuscript and thoroughly checked the revised manuscript to avoid any grammatical errors.
Round 2
Reviewer 1 Report
Overall, I am satisfied, although I still think validation of transcriptomic data for at least a few genes would be very useful. I hope that these important data will be published by the authors in their next article.
Author Response
Thank you very much for your suggestion. As we indicated last time, we are working on the pathway analysis and will certainly incorporate the details in our next article.
Reviewer 2 Report
- In Figure 6, the autors staed that Tumor Growth Factor pathway (TGF) has components as Smad and BMP. Actually, the TGF-β/BMP pathway that was also indicated in abstract reffers to transforming growth factor-β.
Author Response
Our sincere apologies for the mishap. It is indeed Transforming Growth Factor. We have changed the erroneous annotation in figure 6 (Tumor Growth Factor pathway) to Transforming Growth Factor pathway.